# Multi-convolutional neural networks for cotton disease detection using synergistic deep learning paradigm

Afira Aslam[1], Syed Muhammad Usman[2], Muhammad Zubair[3], Amanullah Yasin[2], Muhammad Owais[4]*, Irfan Hussain[4]*

1 Department of Creative Technologies, Faculty of Computing and Artificial Intelligence, Air University, Islamabad, Pakistan, 2 Department of Computer Science, Bahria School of Engineering and Applied Sciences, Bahria University, Islamabad, Pakistan, 3 Interdisciplinary Research Center for Finance and Digital Economy, King Fahd University of Petroleum and Minerals, Dhahran, Saudi Arabia, 4 Department of Mechanical and Nuclear Engineering, Khalifa University, Abu Dhabi, United Arab Emirates

* muhammad.owais@ku.ac.ae (MO); irfan.hussain@ku.ac.ae (IH)

**Data availability statement:** The data underlying the results presented in the study are available from the following sources: (1) Kaggle:

## Abstract

Cotton is a major cash crop, and increasing its production is extremely important worldwide, especially in agriculture-led economies. The crop is susceptible to various diseases, leading to decreased yields. In recent years, advancements in deep learning methods have enabled researchers to develop automated methods for detecting diseases in cotton crops. Such automation not only assists farmers in mitigating the effects of the disease but also conserves resources in terms of labor and fertilizer costs. However, accurate classification of multiple diseases simultaneously in cotton remains challenging due to multiple factors, including class imbalance, variation in disease symptoms, and the need for real-time detection, as most existing datasets are acquired under controlled conditions. This research proposes a novel method for addressing these challenges and accurately classifying seven classes, including six diseases and a healthy class. We address the class imbalance issue through synthetic data generation using conventional methods like scaling, rotating, transforming, shearing, and zooming and propose a customized StyleGAN for synthetic data generation. After preprocessing, we combine features extracted from MobileNet and VGG16 to create a comprehensive feature vector, passed to three classifiers: Long Short Term Memory Units, Support Vector Machines, and Random Forest. We propose a StackNet-based ensemble classifier that takes the output probabilities of these three classifiers and predicts the class label among six diseases—Bacterial blight, Curl virus, Fusarium wilt, Alternaria, Cercospora, Greymildew—and a healthy class. We trained and tested our method on publicly available datasets, achieving an average accuracy of 97%. Our robust method outperforms state-of-the-art techniques to identify the six diseases and the healthy class.

Cotton Leaf Disease Dataset (https://www.kaggle.com/datasets/seroshkarim/cotton-leaf-disease-dataset); (2) Mendeley Data: Cotton Plant Disease Dataset (https://data.mendeley.com/datasets/6hm6pc3y43/2); (3) Roboflow: Cotton Plant Disease Prediction Dataset (https://universe.roboflow.com/national-college-of-ireland/cotton-plant-disease-prediction-igthk/dataset/3); (4) Roboflow: Cotton Plant Disease Dataset (https://universe.roboflow.com/roboflow-100/cotton-plant-disease/dataset/2).

**Funding:** This work was supported by KUCARS, Department of Mechanical and Nuclear Engineering, Khalifa University under Award numbers: FSU-2021-019 and RC1-2018-KUCARS.

**Competing interests:** The authors have declared that no competing interests exist.

## Introduction

Agriculture is the primary industry for growing plants, harvesting crops, and raising livestock to produce extra food for the world's population. Crops, dairy products, edibles, and other main agricultural products are essential to daily living [1]. A little over 40% of the world's population work in agriculture. Fewer people are employed in agriculture now than in the past [2]. Cotton yield in some of the major countries is shown in Table 1.

Artificial intelligence (AI) is a type of automation used in agriculture to increase productivity, handle difficulties, and find solutions to issues in diverse crop fields [3]. AI can predict crop yield by incorporating technology to collect data on leaf diseases, soil moisture, meteorological conditions, pest infestations, and crop growth. With AI, employing robots, drones, and sensors in agriculture, agronomists can generate high-quality images of crops, increasing the detection output. AI's main applications in farming are its versatility, performance, accuracy, and cost-effectiveness [4].

The agriculture datasets have been broken down into smaller pieces, and their trends and behaviors have been analyzed for handling a large volume of data [5]. AI should be able to handle agriculture datasets using machine learning and deep learning [28] domains that improve the machines and increase prediction accuracy. Crop disease is a crucial problem in agriculture since it significantly damages crops [6]. The biggest threat to agriculture is crop disease, which reduces yields and the quality and amount of food produced. Cotton is an important economic crop that aids in creating natural fiber. It promotes the expansion of the textile sector. Fig 1 shows seven diseases in cotton plants that are commonly found. These diseases affect the yield of the cotton crop, which causes heavy losses to the farmers.

AI offers a way to determine plant diseases precisely. Deep learning is an automated approach that helps precisely and accurately recognize agricultural diseases while saving money and time. Early and precise recognition of damaged cotton plant leaves is essential in computer vision because of the disease and the changing environment. The complexity is increased by the image's angle, background, and noise in it. The visual symptoms help in the identification of the disease. Contrarily, it is difficult to create an image with a complex background [7]. The damaging biological risks, such as diseases and pests during the cotton-growing crop times, cost agronomists a tremendous amount of money. Deep learning systems can classify and forecast the damaged cotton crop. Images of the cotton crop are subjected to image processing for segmentation, classification, and identification to provide agricultural scientists with high-quality data. The deep learning algorithms use the segmented image to distinguish between healthy and diseased leaves.

**Table 1. Cotton production in major countries.**

| Major countries | Production, in thousand metric tons |
|---|---|
| China | 5879 |
| India | 5334 |
| United States | 3815 |
| Brazil | 2678 |
| Pakistan | 1306 |
| Australia | 1197 |
| Turkey | 827 |
| Uzbekistan | 577 |
| Argentina | 327 |
| Mali | 311 |

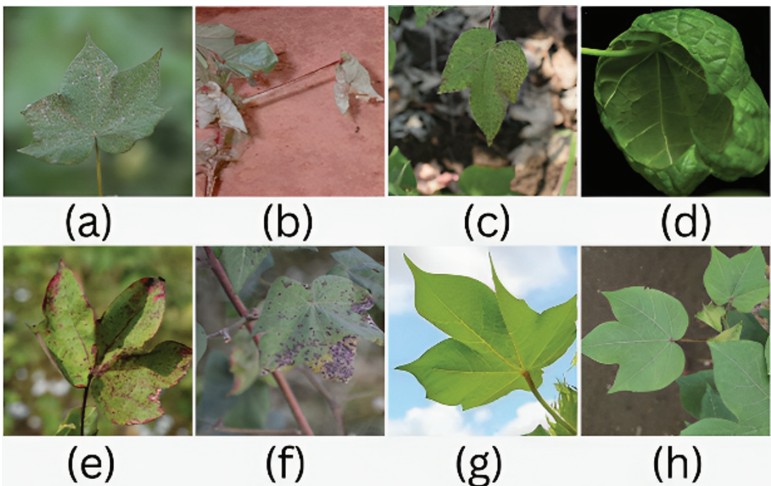

**Fig 1. Representative sample images from each category used in the multi-class classification task.** (a) Grey mildew, (b) Fusarium wilt, (c) Cercospora, (d) Curl virus, (e) Bacterial blight, (f) Alteranaria, (g–h) Healthy.

## Key contribution

A robust deep learning system is essential for addressing the practical challenges encountered in real-world agricultural settings. These challenges include diverse symptoms of diseases, such as varying sizes and colors of lesions, which can complicate accurate diagnosis. Additionally, occlusion, where leaves are partially covered by other leaves or objects, poses a significant obstacle to clear image capture and analysis. Variations in leaf appearance, including differences in shapes, sizes, and textures, further add to the complexity. To tackle these issues, we proposed a comprehensive system to manage and overcome these challenges effectively, ensuring reliable and accurate performance in diverse agricultural environments. Contributions of this research to the body of knowledge include:

- Developed a novel method for data augmentation by combining conventional methods and custom Style GANs to reduce class imbalance problems.
- Introduced feature-level fusion scheme by combining features extracted from MobileNet and VGG16 to create a comprehensive feature vector
- Proposed an ensemble classifier using StackNet by integrating Long Short Term Memory Units (LSTM), Support Vector Machine (SVM), and Random Forest to classify cotton crop diseases accurately.
- Achieved an average accuracy of 97% on publicly available datasets, surpassing state-of-the-art techniques in classifying seven classes, including six diseases and a healthy class.

The rest of the paper is organized as follows: the section Literature review provides the related work. The proposed methodology is presented in the section Proposed Methodology and contains a dataset description, whereas results are presented by varying experimental steps in the section Result. A detailed discussion comparing the state of the art is presented in the section Discussion, and the final section Conclusion concludes this research and presents future directions.

## Literature review

A typical method for automated disease detection involves four steps. First is data acquisition, which can be performed using a camera mounted in the field, drones, or a mobile device. After data acquisition, the next step is to preprocess the data to remove the noise and mitigate the class imbalance problem. In the third step, a feature vector is extracted that distinguishes between multiple classes, followed by the classification, which is the last step. Researchers have recently proposed multiple methods for automated disease detection on cotton crops using machine and deep learning models. Table 2 provides a critical analysis of the state-of-the-art methods proposed in recent years for classifying diseases on cotton lead. Deep learning was used by Singh et al. [9] and Devi et al. [10] to improve plant disease classification. Researchers [9,11] used the ResNet152V2 model, accomplishing an astounding 97% categorization accuracy. However, Devi et al. used ELM to explore the SSADN-PLDDC technique and achieved an impressive 97.87% accuracy. Few researchers [10] neglected a more thorough examination of how hyperparameters affected the effectiveness of the ELM. Combining the two strategies and utilizing the strength of ResNet-based designs with

**Table 2. Comparison of state-of-the-art methods proposed by researchers in recent years for automated detection of cotton diseases.**

| Title | Year | Dataset | Method | Accuracy | Limitations |
|---|---|---|---|---|---|
| Nazeer et al. [46] | 2024 | Kaggle Dataset and proprietary collection of 1349 images | CNN | 99% | Binary classification |
| Parashar et al. [47] | 2024 | Real-time dataset from drones | MobileNetV2, CNN-based model | 99.91% | Focus on diseases caused by pathogens only |
| Wenjuan et al. [48] | 2024 | Self-built cotton leaf disease dataset | Improved SSD (Single Shot Multi-Box Detector) combined with MobileNetV2 | 3.8% increase | Binary dataset, low performance metric |
| Rai et al. [49] | 2023 | Self-built cotton leaf disease dataset | CNN | 97.98% | Classification on 4 types |
| Shrivastava et al. [50] | 2023 | Self-built cotton leaf disease dataset | CNN | 99.67% | Spread of disease is not discussed |
| Singh et al. [9] | 2023 | Plant Village Dataset | ResNet152V2 | 97% | Segmentation of disease is not covered |
| Mohanavel et al. [12] | 2022 | Very high-resolution images from 450km above | Modified Classifier Logic for Crop Monitoring | 85% | Combine mapping, machine-to-machine (M2M), and cloud computing technologies |
| Devi et al. [10] | 2022 | Dataset on plant diseases from Kaggle repository | SSADN-PLDDC technique with extreme learning machine (ELM) | 97.87% | Limited dataset, Lack of analysis on the effect of hyperparameters |
| Noon et al. [16] | 2021 | Cotton leaf disease publicly available | EfficientNet and MobileNet model | 99.95% | Limited data, Lack of analysis on the effect of preprocessing techniques, and only 4 classes were considered |
| Zekiwos et al. [33] | 2021 | Primary dataset is self-collected, and secondary dataset is obtained from MelakaWorker agricultural research center | CNN | 96.40% | Effectiveness of various hyperparameters on the suggested deep learning model's performance not examined, did not compare with state-of-the-art models |
| Patil et al. [2] | 2021 | Every image was gathered via a daily survey using an IoT-based system camera and sensor installed in a crop field | CNN | 98% | The proposed methodology for the detection of diseases in the cotton plant is time-consuming |
| Ye et al. [34] | 2020 | 120 samples collected from Guangxi site and Hainan site | Binary logistic regression | 80% | Examining the variations in spectral response characteristics in Fusarium wilt and other yellowing phenomena |
| Pechucho et al. [35] | 2020 | ImageNet dataset | DNN inception-V3 pre-trained model | 90% | Multiple crops can be detected using the same methods; this study can be extended to find fair use of pesticides after disease detection |
| Shakeel et al. [36] | 2020 | Public dataset on Cercospora Leaf Spot | K mean clustering and SVM | 96% | Size of dataset can be increased; work is only focused on a single disease |

meticulous hyperparameter optimization to maximize the potential of the ELM presents an exciting opportunity.

Mohanavel et al. [12] used high-altitude photos taken from an astounding 450 km above the planet's surface to monitor crops. With a changed classifier logic, they achieved an accuracy of 85%. This method represents a paradigm shift from traditional ground-based evaluations and shows the promise of remote sensing in disease identification. This technology was expanded by Wang et al. [13] by using unmanned aerial vehicles to obtain high-resolution photos. Their accuracy rate with the KMSEG classifier was 88.39%. The synergy between these investigations emphasizes how remote sensing has the potential to revolutionize disease detection. Nonetheless, a comprehensive assessment methodology that includes preprocessing methods, a range of evaluation measures, and implications for practical applicability would benefit both investigations.

With the help of the Deep Neural Networks Inception-V3 pre-trained model and the ImageNet dataset, Pechucho et al. [14] obtained an impressive 90% accuracy. Their innovative proposal to expand their system to identify other crops demonstrates the possibility of a single strategy in various agricultural contexts. On the other hand, Shakeel et al. [15] targeted study on Cercospora Leaf Spot detection showed that $k$-mean clustering and SVM could detect the disease with 96% accuracy. Combining these findings could spur the development of an all-encompassing strategy for managing multi-crop diseases. However, both studies could strengthen their findings by working with larger datasets and refining their methodologies to cover a broader range of diseases. Noon et al. [16] and Xavier et al. [17] offer two different strategies for tackling the challenges of incomplete data and identifying multiple diseases. Noon et al.'s [16] work, using EfficientNet and MobileNet models, demonstrates an impressive 99.95% accuracy in diagnosing cotton leaf diseases, showcasing the effectiveness of cutting-edge models in situations with limited data.

Researchers [18,20] have applied multiple methods for the detection and classification of plant diseases, including techniques like the $k$-means algorithm and neural networks, which show good results in the automated detection of diseases on cotton leaves with accuracies between 85% and 89.56%. Moreover, using RGB and HSV components with Artificial Neural Networks also shows an efficient way of disease detection [68]. Preprocessing methods transform color images into grayscale images and also help in disease detection using textural details. Color space segmentation, modified feature maps, Support Vector Machine classification [70], and Gabor filters [71] have also been utilized in literature to detect disease accurately. Dimensionality reduction and scatter matrices are employed to extract key features from images of cotton leaves. ANN [21], along with various image processing techniques, have been used to achieve high accuracy of cotton disease detection. Rothe and Kshirsagar [22] emphasize taking pictures using digital cameras and then post-processing them with Low Pass and Gaussian filters. Seven Hu's moments are used to generate the features for the classifier training, which are then further segmented using an active contour model. Utilizing this feature vector for disease classification allows the feed-forward back propagation method to be applied.

The image processing technique is also used [23–27,29] to provide an automated method for disease diagnosis on cotton leaves. The SVM classifier, which has an accuracy of 98.47%, is used for classification using extracted features such as texture and color. Filtering, background removal, and enhancement are preprocessing operations. Color-based segmentation is used to extract sick segments from cotton leaves. Notably, [30] explores a novel approach by focusing on specific lesions and patches rather than complete leaves. This expands the

dataset without increasing image count, facilitating multi-disease identification on the same leaf. However, manual symptom separation remains necessary, limiting complete automation. Despite this, the method exhibits enhanced accuracy, showcasing the potential of deep learning techniques for plant disease diagnosis when sufficient data is available, even if not fully exhaustive in representing all practical scenarios. The potential of three-band multispectral images to identify ramularia leaf blight is evaluated across altitudes [31]. While infection levels correlated with altitude increase, distinguishing between illness severity levels posed challenges. Despite varying accuracy, the study underscores the promise of low-cost multispectral devices for detecting cotton ramularia blight. Similarly, another study [32] focuses on automated early-stage cotton illness identification through image processing. Employing the $k$-means clustering technique for segmentation, the hybrid approach integrates texture and color feature extraction.

SVM classified Cercospora cotton leaves with an accuracy of 96%. Exploring the role of mobile robots in agriculture [41], a robot designed for crop inspection is constructed, modeled, and controlled similarly to differential-drive vehicles. Successful autonomous navigation is ensured via odometry and cameras, proving effective for electro-mechanical row crop navigation. Addressing precise self-localization challenges in agriculture, Liu's report advocates using an above-ground image coupled with crop and weed semantics. The technology demonstrates the potential for precision agriculture applications. For disease identification, Zekiwos and Bruck [33] utilizes GLCM, thresholding, segmentation, and feature extraction. Digital images of RGB leaf captures are processed, considering color transformation and extraction of texture features. In [34], convolutional neural networks and image processing are harnessed, achieving 96% accuracy for classifying cotton leaf diseases. Researchers [35] focus on disease detection, operating parallel processes for healthy and defective leaf images, employing image manipulation techniques, underscoring the importance of robust training in disease diagnosis.

We have identified multiple research gaps by performing the literature review of cotton leaf disease detection. Firstly, data processing predominantly occurred in controlled environments, neglecting real-world variability. Secondly, limited dataset sizes hindered training deep learning models with robust generalization capabilities. Thirdly, prevalent class imbalances posed challenges to model performance. Fourthly, most deep learning research in disease detection focused on binary or small-scale multi-class classification, neglecting broader disease scenarios. Lastly, exploring disease spread on leaf images to estimate affected areas remains largely unexplored.

Our study addresses the above-highlighted research gaps by introducing a novel approach, which not only contributes to filling the existing void in the literature but also paves the way for future investigations. We developed a novel deep ensemble framework tailored to improve the accuracy of cotton disease classification, successfully addressing the issues of class imbalance and symptom variability. Our approach features a creative data augmentation scheme, which employs a customized StyleGAN and traditional methods such as scaling, rotating, transforming, shearing, and zooming to produce a well-balanced dataset. Additionally, our framework includes a dual-stage fusion process that merges feature sets from MobileNet and VGG16 into a comprehensive feature vector. This vector is then processed by a StackNet-based ensemble classifier combining Long and LSTM, SVM, and RF, achieving an outstanding average accuracy of 97% across two publicly available datasets. This performance significantly exceeds current methods in identifying and classifying six distinct diseases and a healthy condition.

## Proposed methodology

Different machine learning and deep learning models under different experimental settings have been tested to develop an accurate and robust method for disease diagnosis. Fig 2 shows the proposed methodology. It consists of four steps: data acquisition, preprocessing, feature extraction, and classification. The public dataset has been used in this study, and to overcome the problem of data scarcity and imbalance, conventional augmentation techniques, including scale, rotate, shear transform, and zooming, along with StyleGAN, which is a state-of-the-art deep learning model for data augmentation, has been customized and used for data augmentation. In the third step, we propose a multi-convolutional neural network fusion-based feature extraction method that concatenates features obtained from VGG16 and MobileNetV2 by removing the fully connected layers of both architectures.

In the last step, we propose a StackNet meta-learning ensemble model that uses SVM, LSTM, and Random Forest as base models for accurately classifying seven classes, including six diseases and one healthy image class. The use of LSTM, despite its typical application for sequential data, is beneficial as it captures temporal dependencies or patterns within the feature vectors that traditional classifiers like SVM and Random Forest might overlook. By integrating these diverse methods, the strength of LSTM in handling sequential patterns is leveraged alongside the robustness of Random Forest in managing non-linear relationships and the effectiveness of SVM in high-dimensional spaces, thereby enhancing overall model performance. Table 3 presents the proposed architecture of StyleGAN, which is implemented to generate synthetic data to reduce the issue of class imbalance. The use of synthetic data only during training, and highlighting that model evaluation was carried out on real, diverse, and unaltered samples. This helps validate the model's real-world applicability while controlling the effects of any biases introduced by synthetic data.

StyleGAN leverages a series of advanced techniques for generating highly realistic images. The process begins with the transformation of an input latent code $z$, sampled from a normal distribution, into an intermediate latent space $W$ through a mapping network, described by

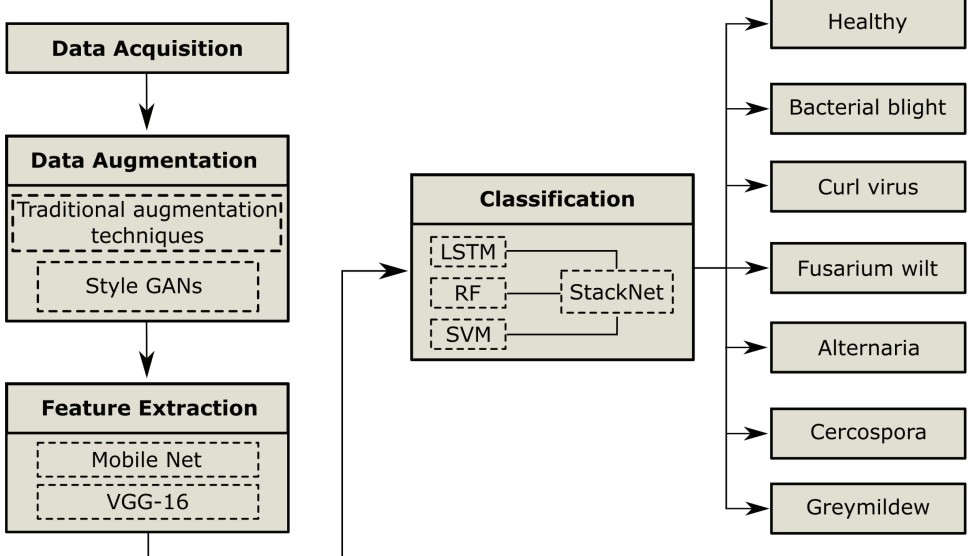

**Fig 2. Flowchart of the proposed model for automated detection of cotton crop disease.**

**Table 3. Architecture of proposed StyleGAN for data augmentation.**

| Component | Output resolution | Number of parameters |
|---|---|---|
| **Generator** | | |
| Mapping network | – | 2,097,152 |
| Constant input | $4 \times 4$ | – |
| Synthesis network | | |
| $4 \times 4$ to $\times 8$ | $8 \times 8$ | 4,194,304 |
| $8 \times 8$ to $16 \times 16$ | $16 \times 16$ | 2,097,152 |
| $16 \times 16$ to $32 \times 32$ | $32 \times 32$ | 1,048,576 |
| $32 \times 32$ to $64 \times 64$ | $64 \times 64$ | 524,288 |
| $64 \times 64$ to $128 \times 128$ | $128 \times 128$ | 262,144 |
| $128 \times 128$ to $224 \times 224$ | $224 \times 224$ | |
| **Discriminator** | | |
| $224 \times 224$ to $128 \times 128$ | $128 \times 128$ | |
| $128 \times 128$ to $64 \times 64$ | $64 \times 64$ | 524,288 |
| $64 \times 64$ to $32 \times 32$ | $32 \times 32$ | 1,048,576 |
| $32 \times 32$ to $16 \times 16$ | $16 \times 16$ | 2,097,152 |
| $16 \times 16$ to $8 \times 8$ | $8 \times 8$ | 4,194,304 |
| $8 \times 8$ to $4 \times 4$ | $4 \times 4$ | 8,388,608 |
| Fully connected layer | 1 | 4,096 |

$W = f(z)$, where $f$ represents the mapping function implemented via fully connected layers. StyleGANs address the problem of data scarcity by generating high-quality synthetic images that are indistinguishable from real images. This increases the dataset size and ensures diversity and variety in the augmented data, thus enhancing the model's ability to generalize better. A key feature, Adaptive Instance Normalization (AdaIN), is applied as

$$\text{AdaIN}(x, y) = y_s \frac{x - \mu(x)}{\sigma(x)} + y_b,$$

adjusting the style of the generated image, where $y_s$ and $y_b$ are scale and bias parameters from the style vector $y$, and $\mu(x)$ and $\sigma(x)$ are the mean and standard deviation of the feature map $x$.

Progressive growing allows the model to start with low-resolution images, adding layers to increase resolution and facilitating a coarse-to-fine learning approach. The truncation trick, modulating the latent space $W$ with a parameter $\psi$, is used to adjust the balance between diversity and fidelity, yielding $W' = \psi W + (1 - \psi)\overline{W}$, with $\overline{W}$ as the average style vector.

To introduce stochastic variations, noise $n$ is added to feature maps $x$, resulting in $x' = x + n \cdot \alpha$, where $\alpha$ is a per-feature learned scaling factor.

The GAN loss function, optimizing the generator $G$ and discriminator $D$, is formulated as $\mathcal{L}_{GAN}(G, D) = \mathbb{E}_{z \sim p_z}[\log(1 - D(G(z)))] + \mathbb{E}_{x \sim p_{data}}[\log D(x)]$, where $z$ and $x$ represent latent vectors and real images, respectively.

The VGG16 architecture, renowned for its simplicity and depth, significantly influenced convolutional neural network designs. Central to VGG16 is its uniform use of $3 \times 3$ convolutional layers stacked sequentially to capture complex patterns within images. The convolution operation mathematically represents this design choice:

$$f_{\text{out}} = \text{ReLU}(W * f_{\text{in}} + b) \tag{1}$$

**Table 4. VGG16 Architecture with detailed list of parameters for automated feature extraction.**

| Layer Type | Output Size | Parameters |
|---|---|---|
| Input | $224 \times 224 \times 3$ | 0 |
| Conv3x3 + ReLU (x2) | $224 \times 224 \times 64$ | 38,720 |
| MaxPooling | $112 \times 112 \times 64$ | 0 |
| Conv3x3 + ReLU (x2) | $112 \times 112 \times 128$ | 221,440 |
| MaxPooling | $56 \times 56 \times 128$ | 0 |
| Conv3x3 + ReLU (x3) | $56 \times 56 \times 256$ | 1,475,328 |
| MaxPooling | $28 \times 28 \times 256$ | 0 |
| Conv3x3 + ReLU (x3) | $28 \times 28 \times 512$ | 5,899,776 |
| MaxPooling | $14 \times 14 \times 512$ | 0 |
| Conv3x3 + ReLU (x3) | $14 \times 14 \times 512$ | 7,079,424 |
| MaxPooling | $7 \times 7 \times 512$ | 0 |
| Fully Connected + ReLU | 4096 | 102,764,544 |
| Fully Connected + ReLU | 4096 | 16,781,312 |
| Fully Connected + Softmax | 1000 | 4,097,000 |

where $f_{\text{in}}$ and $f_{\text{out}}$ are the input and output feature maps, $W$ represents the weights of the $3 \times 3$ convolutional filters, $b$ is the bias, and ReLU denotes the Rectified Linear Unit activation function.

Following each convolutional block, VGG16 employs max pooling to reduce spatial dimensions, thereby condensing information and reducing computation for subsequent layers:

$$f_{\text{reduced}} = \text{MaxPool}(f_{\text{out}}) \tag{2}$$

The network culminates in fully connected layers, designed to flatten the high-level features extracted by the convolutional layers into a vector, which is then mapped to the desired number of classes for classification tasks. Table 4 presents the architecture and list of parameters of the VGG16 used in this research for feature extraction.

MobileNetV2 introduces the concept of depth-wise separable convolutions and a residual structure. Due to this, it provides high accuracy with less computational cost. Depth-wise separable convolution consists of depthwise convolution followed by point-wise convolution.

$$f_{\text{depth-wise}} = \text{depth-wiseConv}(f_{\text{in}}) \tag{3}$$

$$f_{\text{pointwise}} = \text{Conv1x1}(f_{\text{depth-wise}}) \tag{4}$$

It applies a single filter per channel, and pointwise convolution ($1 \times 1$ convolution) combines the output of the depth-wise convolution across multiple channels. This operation enables the use of fewer parameters.

The inverted residual block of MobileNetV2 consists of an initial expansion layer ($1 \times 1$ convolution), a depth-wise convolution. Then, a projection layer ($1 \times 1$ convolution) is used to condense the feature maps, interspersed with shortcut connections. This design optimizes information flow and efficiency:

$$f_{\text{expanded}} = \text{Conv1x1}(f_{\text{in}}, \text{expansion factor}) \tag{5}$$

$$f_{\text{projected}} = \text{Conv1x1}(\text{ReLU6}(\text{depth-wiseConv}(f_{\text{expanded}}))) \tag{6}$$

Table 5 details the trainable parameters required in MobileNet V2. MobileNetV2 is a lightweight architecture that classifies images in real-time mobile devices with low computational complexity. Fig 3 shows the layer-wise diagram of the MobileNetV1. MobileNetV2 builds on the original MobileNetV1 using depth-wise separable convolutions to reduce computational cost and achieve high accuracy. Depth-wise convolution and point-wise convolution are the main reasons behind the low computational cost of MobileNetV2.

$$Y = F(X) + X_1 \tag{7}$$

where $Y$ denotes the output, $F(X)$ the convolutional process, and $X$ the input feature map. We have combined features extracted from VGG16 and MobileNetV2 to form a single feature vector, which is then fed as input to the three different classifiers, including LSTM, RF, and SVM. The output of these classifiers is combined using StackNet to get the final output. The ensemble classification method improves the accuracy of disease detection. The process of ensemble classification using Stacknet is as follows:

$$F = \text{MetaClassifier}\left(\sum_{i=1}^{n} w_i \cdot O_i\right) \tag{8}$$

**Table 5. MobileNetV2 architecture with detailed list of parameters for automated features extraction.**

| Layer type | Output size | Parameters |
|---|---|---|
| Input | $224 \times 224 \times 3$ | 0 |
| Conv3x3 + ReLU6 | $112 \times 112 \times 32$ | 864 |
| Bottleneck (x1) | $112 \times 112 \times 16$ | 5,136 |
| Bottleneck (x2) | $56 \times 56 \times 24$ | 28,224 |
| Bottleneck (x3) | $28 \times 28 \times 32$ | 47,840 |
| Bottleneck (x4) | $14 \times 14 \times 64$ | 88,064 |
| Bottleneck (x3) | $14 \times 14 \times 96$ | 137,088 |
| Bottleneck (x3) | $7 \times 7 \times 160$ | 320,160 |
| Bottleneck (x1) | $7 \times 7 \times 320$ | 472,064 |
| Conv1x1 + ReLU6 | $7 \times 7 \times 1280$ | 409,600 |
| Average Pooling | $1 \times 1 \times 1280$ | 0 |
| Fully Connected + Softmax | 1000 | 1,280,000 |

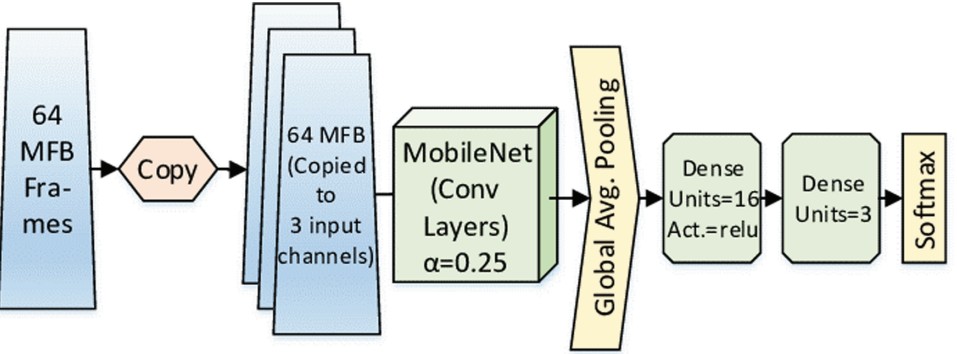

**Fig 3. Architecture of the MobileNet proposed for feature extraction.**

In the above-mentioned equation, *F* is the output produced by the StackNet, which is the final classification result. The ensemble classifier effectively integrates these inputs to produce the final decision and improves overall classification accuracy.

Meta has launched a new "Segment Anything" project to create two crucial parts. The Segment Anything paradigm (SAM) [43] is a foundation paradigm for quick picture segmentation. This draws inspiration from the field of NLP, where big datasets (worth billions of tokens) and foundation models are becoming standard. The project results in the simultaneous development of a sizable dataset, a segmentation model, and a data engine. The scientists picked picture segmentation as the starting point for these enormous models and datasets since it is one of the fundamental computer vision problems. Image segmentation offers a variety of possible applications in both science and artificial intelligence. Fig 4 contains segment anything models general framework.

The proposed methodology employs pre-trained deep-learning models to distinguish between healthy and diseased cotton leaves and identify the disease type from leaf images. Machine learning classifiers are utilized to extract the features of VGG16 and MobileNetV2. The classifiers are being trained to categorize images into disease groups, with performance metrics being used to compare and identify the optimal classifier. A "Segment Anything Model" is used for precise pixel-level segmentation of diseased areas, aiding in calculating disease-covered area percentage.

We have trained the model by changing hyperparameters, including different numbers of epochs, optimizers, activation functions, number of neurons in a dense layer, and ensemble classification. MobileNetV2 performs well and achieved test accuracies between 0.92 and 0.97, whereas, DenseNet121 achieved 0.89 test accuracy. Although results from other models vary, this research points out that MobileNetV2 and DenseNet121 show promise as powerful options for identifying and mapping out diseases on cotton leaves. After performing an ablation study, the proposed method was finalized. The proposed method is compared with the state-of-the-art existing methods for cotton disease detection, and the proposed method performs well in terms of both the number of classes and accuracy. The proposed method achieves a testing accuracy of 0.97 with seven classes, including six diseases and a healthy class.

Jupyter BBox is used to annotate the images for the SAM model for pixel-level segmentation to get the area affected by the disease. First, we bring in the required libraries and then

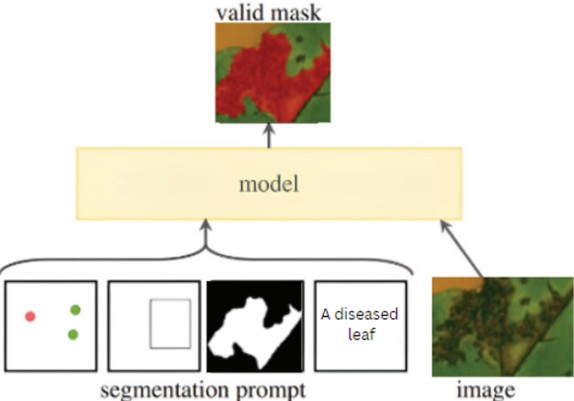

**Fig 4. Segment Anything Model (SAM) framework for segmentation of cotton crop leaves.**

create a BBoxWidget. This lets users outline the areas they're interested in with bounding boxes. Next, we load the mask predictor and feed the image into it, using the MaskPredictor's predict() method. This process gives us the expected masks, scores, and logits. A BoxAnnotator and a MaskAnnotator are generated to annotate identified objects with bounding boxes and segmentation masks. A detection object is formed from the predicted masks, and the object with the most significant area is selected.

The source and segmented images are then created using the BoxAnnotator and MaskAnnotator's methods. SAM effectively segments various disease-affected leaf images from controlled environments or field conditions. Using an annotated dataset, SAM can aid disease classification from leaf images with reduced resource and time requirements. This underscores SAM's potential for practical disease classification applications. Fig 5 shows the source and segmented images, and Fig 6 shows multiple bounding boxes.

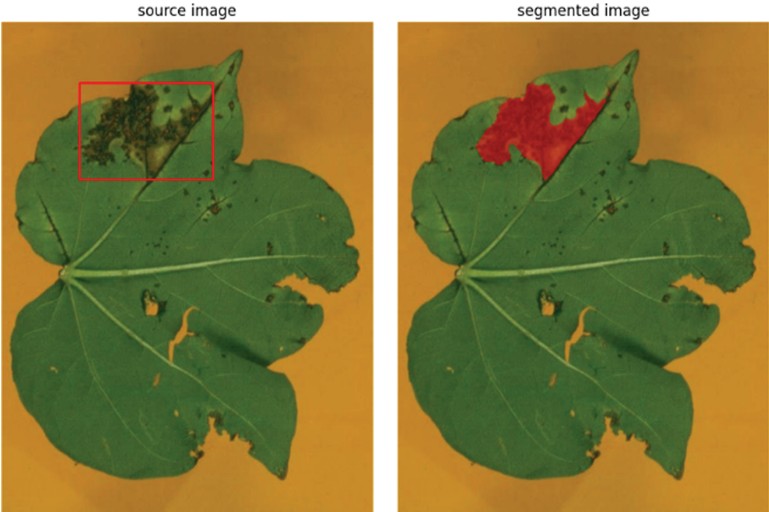

**Fig 5. Segmentation results obtained from SAM on cotton leaf image.**

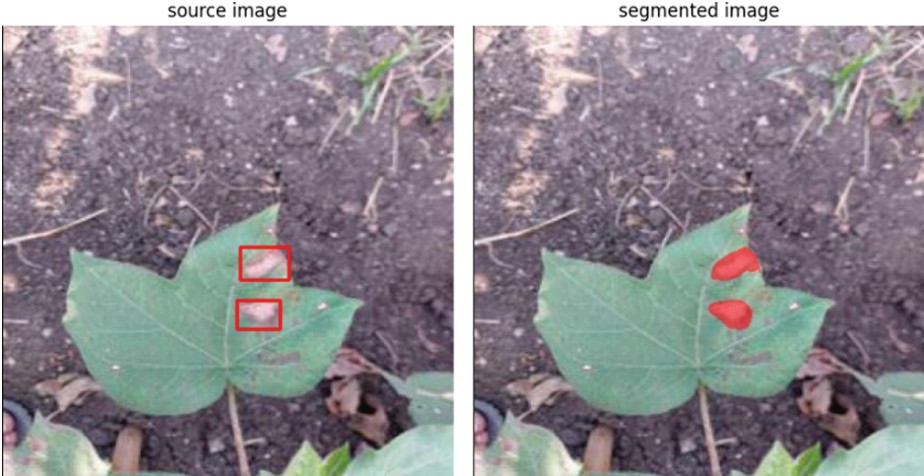

**Fig 6. Results of multiple segmentation on single leaf image using SAM.**

For feature extraction, a machine learning pipeline extracts features from images using a pre-trained VGG16 and MobileNetV2, trains different machine learning models on the collected features, and evaluates the classifier's performance on a test set. The first section of the code imports necessary libraries, including Keras, which is used to load the VGG16 and MobileNetV2 models. Next, the VGG16 model is loaded with pre-trained weights from the ImageNet dataset. In subsequent steps, this model is utilized to extract features from images.

A function named "extract_features" processes images, applying Keras utilities, VGG16, and MobileNetV2 models. Features and labels are flattened, stored in arrays, and saved as numpy arrays. These are used for training and testing sets. Logged through Scikit-learn's Pipeline and GridSearchCV, machine learning models are evaluated using hyperparameter tuning with a parameter grid. Grid search cross-validation assesses performance, and accuracy is measured using Scikit-learn's score function. StackNet—a powerful ensemble learning model, is used later to classify diseases. StackNet excels in combining diverse base models, enhancing predictive performance by leveraging their unique strengths. The combined predictions are calculated by linearly combining base models along with their weights, as shown in equation 2.

$$P\_ensemble(X) = w_1.Pm_1(X) + w_2.Pm_2(X) + .... + w_k.Pm_k(X), \qquad (9)$$

where P_ensemble(X) represents the meta-model prediction and $m_1, ... m_k$ represents base models. The loss function is

$$\sum_{i=1}^{N} loss(Y_i, P\_ensemble(x_i)), \qquad (10)$$

## Results

### Dataset and experimental setup

This research has used two publicly available datasets [44,45]. The combined dataset used in this study includes seven classes, out of which six classes are diseases and one is healthy. The diseases represented in the dataset cover a diverse range of symptoms and appearances, providing a comprehensive basis for evaluating the performance of our classification models. Class imbalance and limited data availability against many classes are challenges. To address the class imbalance issue, we first performed conventional data augmentation techniques such as rotation, scaling, flipping, and cropping. We also employed StyleGAN, a state-of-the-art generative adversarial network, to generate synthetic images. Combining these synthetic images with the real images helped reduce the class imbalance problem. Table 6 provides several images in each class. We split the images into 80,10 and 10 ratios for train, validation, and testing.

### Ablation study

Table 7 presents the accuracy achieved using DT, *k*NN, LSTM, RF, SVM, and an Ensemble Classifier (StackNet) with feature vector extracted from VGG16 and MobileNetV2. Decision Tree classifier shows low performance with MobileNetV2 features (62% to 56%), while KNN demonstrates a significant improvement (65% to 87%). The LSTM classifier performs well with both feature sets, showing an improvement from 84% to 87%, and the Random Forest classifier also improves accuracy from 80% to 83%. The SVM classifier maintains high accuracy, improving slightly from 86% to 88%. The Ensemble Classifier (StackNet) outperforms

**Table 6. Information of dataset with number of images in each class for training, validation, and testing.**

| Image class | Training | Validation | Testing |
|---|---|---|---|
| Greymildew | 832 | 104 | 104 |
| Fusarium wilt | 776 | 97 | 97 |
| Cercospora | 784 | 98 | 98 |
| Curl virus | 776 | 97 | 97 |
| Bacterial blight | 744 | 93 | 93 |
| Alternaria | 776 | 97 | 97 |
| Healthy | 792 | 99 | 99 |

**Table 7. Accuracy of multiple classifiers with a deep-learning-based feature set**

| Classifier | Accuracy (VGG_16 features | Accuracy (MobileNetV2 features) |
|---|---|---|
| Decision Tree (DT) | 0.62 | 0.56 |
| k Nearest Neighbor (KNN) | 0.65 | 0.87 |
| Long Short-Term Memory Units (LSTM) | 0.84 | 0.87 |
| Random Forest (RF) | 0.8 | 0.83 |
| Support Vector Machine (SVM) | 0.86 | 0.88 |
| **Ensemble Classifier (StackNet)** | **0.97** | |

all individual classifiers by a significant margin, achieving a consistently high accuracy of 97% with both feature sets. The percentage gain of the Ensemble Classifier over the best individual classifier (SVM) is approximately 12.79% for VGG16 features and 10.23% for MobileNetV2 features, highlighting the robustness and effectiveness of the proposed method in integrating multiple model predictions.

Fig 7 shows the confusion matrix of the proposed ensemble classifier (StackNet), demonstrating its superior performance and reliability. StackNet combines the output of different classifiers that results in achieving high accuracy with low false positive rate. Confusion matrix shows that the ensemble classification performs better.

The disease portion can be extracted by removing the healthy portion of the image, which is around 17%. Images are segmented into two regions, i.e., healthy and diseased, to compute the area under disease. It is done by pixel-level segmentation and counting the total number of pixels that are affected by the disease. This helps in estimating the severity of the disease. Fig 8 shows the leaf image segmented using the SAM. It identifies the diseased portion of the leaf, which is then highlighted to estimate the disease-affected area.

## Comparison with state of the art methods

The results are derived from different datasets, including Plant Village [59] and Roboflow [60], with either binary or multiclass classification. Among the listed models, DenseNet-121 achieves the highest accuracy (0.99) on the Plant Village dataset [59], followed by MobileNetV2 (0.91), also trained on the same dataset. The proposed ensemble model (MobileNetV2 + VGG16 + StackNet) achieves the third-highest accuracy (0.97), but it is the only approach that integrates multiple datasets (Plant Village + Roboflow), demonstrating its robustness in handling a more diverse dataset and a 7-class classification problem. Other notable models include MobileNetV2 with additional dense layers (0.98) and Xception (0.94), utilizing the Plant Village [59] dataset for multiclass classification. While the proposed method has a higher parameter count (143.4M) compared to lightweight models such as MobileNetV2 (3.5M) and DenseNet-121 (8.0M), its ability to generalize across multiple datasets justifies its complexity. Studies utilizing the Roboflow dataset [60] have reported

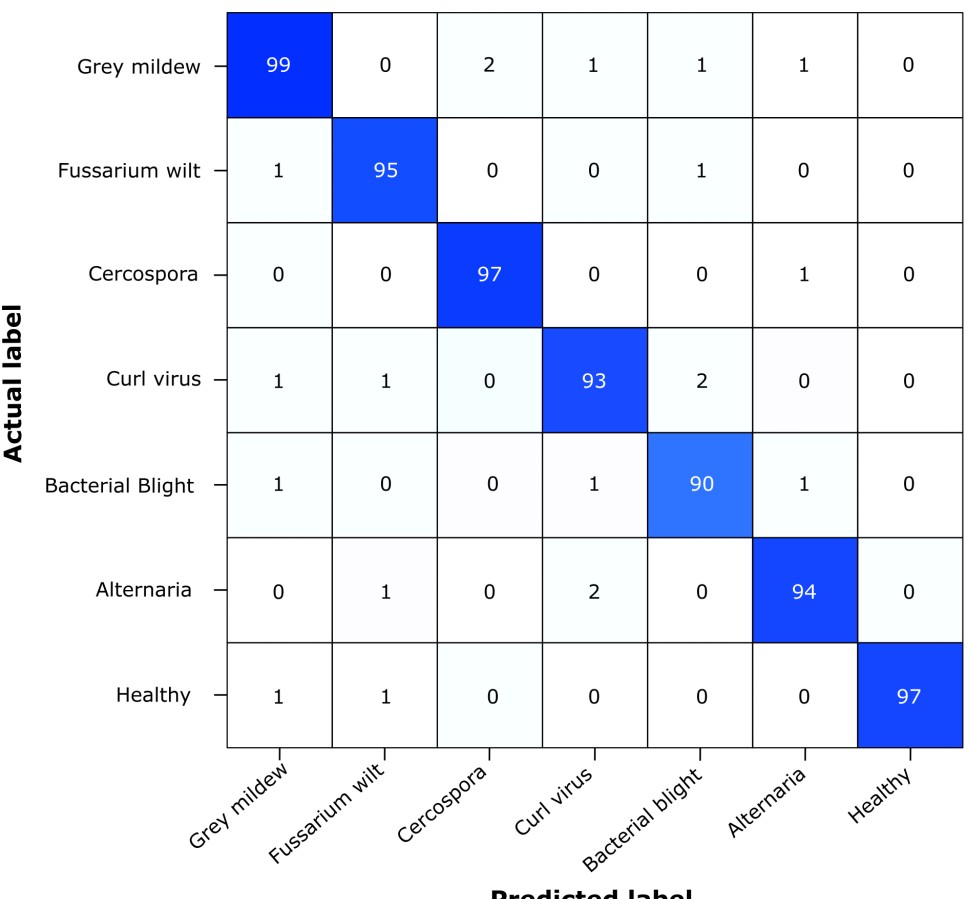

**Fig 7. Confusion matrix for the proposed ensemble classifier.**

an accuracy of 0.96 using a position attention-based capsule network [73] for multi-class classification, and 0.93 using a YOLO-based model [75]. In general, the table highlights that most existing methods rely on a single dataset, whereas the proposed ensemble model effectively uses multiple datasets, making it a more robust and scalable solution for real-world cotton disease detection.

Evaluating a model on a combined dataset, as in our study, introduces a broader range of data variability, including differences in image quality, lighting conditions, background noise, and disease severity. This diversity makes the classification task more complex and better simulates real-world scenarios where such variations are common. In contrast, many prior studies were conducted using a single, often curated dataset like PlantVillage, which may not fully capture these variations and can lead to overly optimistic performance metrics. By training and testing our model on a combination of the PlantVillage and Roboflow datasets, we ensured that it was exposed to a more heterogeneous set of samples. This approach serves as a more rigorous evaluation of the model's robustness, generalization ability, and adaptability to diverse input conditions. Notably, the Roboflow dataset is widely recognized for its greater complexity due to less uniform image characteristics, making it a more challenging benchmark. Our model's superior performance on this dataset, compared to recent state-of-the-art methods, underscores its practical utility and reliability in real-world applications.

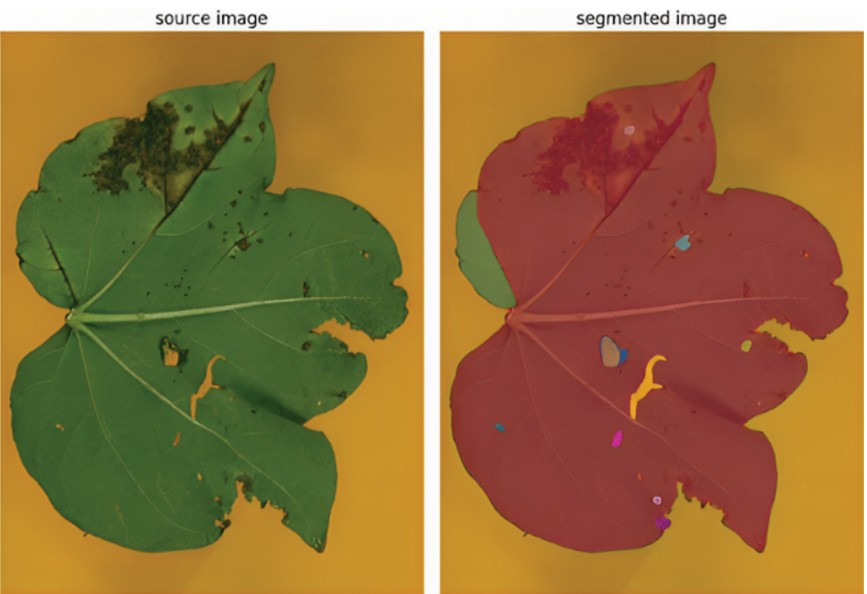

**Fig 8. The whole leaf image segmented using SAM.**

## External dataset testing

To further substantiate the robustness and generalizability of our proposed model, a publicly available cotton leaf disease dataset [76] is used as a benchmark for external testing and independent evaluation. This dataset provides a rich, real-world representation of disease variability and environmental diversity, making it ideal for validating deep learning-based classification models in agricultural domains. The dataset comprises a total of 2137 original images along with 7000 augmented images, designed to enhance training efficacy for deep learning models. These images span seven classes, representing both healthy and diseased cotton leaves across different conditions, including Bacterial blight (250 images), Cotton curl virus (431 images), Herbicide growth damage (280 images), Leaf hopper Jassids (225 images), Leaf reddening (578 images), Leaf variegation (116 images), and Healthy leaf (257 images). The dataset is available at https://data.mendeley.com/datasets/b3jy2p6k8w/2

Each image captures critical disease-specific features such as leaf discoloration, curling, wilting, necrosis, and other symptomatic indicators. The dataset is particularly valuable as it includes images collected from real field environments during different growth stages of the cotton plant. These were taken under varying lighting conditions, natural backgrounds, and imaging devices, reflecting the type of noise and variability encountered in practical settings. The image resolutions vary, including high-resolution formats such as $3000 \times 4000$ pixels, $2239 \times 2239$ pixels, and $1597 \times 1597$ pixels, which contributes to a heterogeneous and challenging dataset for model evaluation. All images were captured through systematic field surveys conducted under expert supervision, ensuring both the accuracy and representativeness of disease labeling. This dataset not only serves as a critical resource for external validation but also acts as a robust benchmark to assess a model's performance beyond controlled training environments. By evaluating our model on this external dataset, we can verify its ability to generalize well across unseen and more complex data distributions, thereby supporting its practical applicability in precision agriculture.

**Table 8. Comprehensive comparative analysis of the proposed model versus existing methods for cotton disease detection and other applications in the agricultural domain. Results are based on different PlantVillage [59] and Roboflow [60] datasets with multiclass classification.**

| Author | Year | Model | #Par (M) | Accuracy | Dataset | Classification |
|---|---|---|---|---|---|---|
| Wang et al. [54] | 2023 | MobileNetV2 | 3.5 | 0.91 | Plant village [59] | Multiclass |
| Wang et al. [54] | 2023 | MobileNetV2 + more dense layers | 4 | 0.98 | Plant village [59] | Multiclass |
| Alam et al. [53] | 2024 | Xception | 22.9 | 0.94 | Plant village [59] | Multiclass |
| Alam et al. [53] | 2024 | Resnet152 | 60.4 | 0.93 | Plant village [59] | Multiclass |
| Alam et al. [53] | 2024 | EfcientNetB4 | 19.3 | 0.96 | Plant village [59] | Multiclass |
| Eunice et al. [55] | 2022 | DenseNet-121 | 8.0 | 0.99 | Plant village [59] | Multiclass |
| Bhatheja et al. [57] | 2021 | Deep CNN | 5.3 | 0.90 | Plant village [59] | Multiclass |
| Bhujade et al. [73] | 2025 | Position attention-based capsule network | 6.2 | 0.96 | Roboflow [60] | Multiclass |
| Maqbool et al. [75] | 2023 | YOLO-based Object Detection Model | 7.2 | 0.93 | Roboflow [60] | Multiclass |
| **Proposed Method** | - | Ensemble (MobileNetV2+VGG16+StackNet) | 143.4 | **0.97** | Plant village [59]+Roboflow [60] | Multiclass (7 classes) |

Table 9 presents a comparative analysis of our proposed model's performance against several recent state-of-the-art methods on the publicly available cotton leaf disease dataset [76]. Although the accuracy of our model (0.964) is marginally lower than some existing models that achieved up to 0.992, it is important to emphasize that our model was evaluated entirely on unseen data from this dataset. In contrast, the referenced models often utilized subsets of this same dataset for both training and validation, leading to more favorable but potentially biased accuracy outcomes.

The objective of this external validation was not solely to outperform existing methods on this dataset, but rather to test the generalizability and robustness of our model when applied to entirely new data under different conditions. Our model had no prior exposure to this dataset during training, which makes the achieved performance particularly noteworthy. Additionally, our model has already demonstrated strong and consistent results on two diverse datasets, including "PlantVillage" [59] and Roboflow [60], both of which encompass variations in lighting, background, disease severity, and image quality.

The combination of high performance on varied training datasets and consistent accuracy on this unseen external dataset illustrates the robustness and adaptability of our approach. This validates the reliability of the proposed model beyond ideal or curated data environments. Hence, these results strongly support the practical applicability of our model in real-time settings, where data is often heterogeneous and unpredictable. Our model's ability

**Table 9. Performance comparison of the proposed model on the cotton leaf disease dataset [76] against State-of-the-Art methods for robustness evaluation.**

| Author | Year | Model | Accuracy |
|---|---|---|---|
| Bishshash et al. [76] | 2024 | Inception V3 | 0.960 |
| Nirob et al. [77] | 2025 | Atrous Spatial Pyramid Pooling with Squeeze-and-Excitation blocks | 0.992 |
| Patra et al. [78] | 2024 | Modified MobileNet | 0.984 |
| Rahman et al. [79] | 2025 | DenseNet169 with Convolutional Block Attention Module (CBAM) | 0.962 |
| | | EfficientNetB1 with CBAM | 0.992 |
| **Proposed Method** | - | Ensemble (MobileNetV2+VGG16+StackNet) | 0.964 |

to generalize well, even on unseen data, confirms its potential for real-time cotton disease classification in practical agricultural scenarios.

## Visual explanation of model decisions

The interpretability of the deep learning model gain trust among agricultural experts and farmers, so, we incorporated explainable AI techniques, specifically Gradient-weighted Class Activation Mapping (Grad-CAM). Grad-CAM offers visual explanations by highlighting the regions in the input image that the model focuses on while making a classification decision. This provides insights into the inner workings of the model and helps validate whether the decisions are being made for the right reasons. We applied Grad-CAM on correctly classified test samples from the Plant Village [59] dataset and visualized the attention maps generated for VGG-16, MobileNet V2, StackNet, and our proposed ensemble model. As shown in Fig 9,

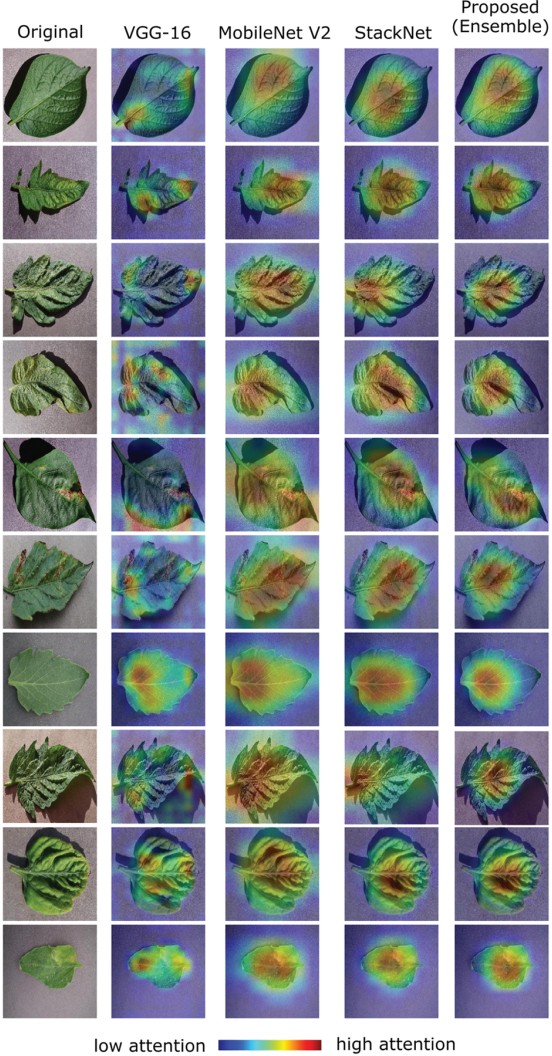

**Fig 9. Grad-CAM visualizations illustrating model interpretability for cotton disease classification on samples from Plant Village dataset [59].** The heatmaps highlight the discriminative regions used by each model to make predictions. The images exhibit attention on relevant symptomatic regions.

the heatmaps reveal that the proposed ensemble model consistently attends to the most relevant symptomatic regions of cotton leaves, such as areas of necrosis, discoloration, or curling, which are key visual indicators for disease identification.

In comparison, the individual models (VGG-16 and MobileNet V2) sometimes highlight broader or less specific regions, which could lead to misinterpretation or reduced confidence in the model's focus. The StackNet, while more refined than individual models, also shows scattered attention in some cases. The ensemble model, by aggregating and refining the strengths of these networks, demonstrates a more precise and localized focus on disease-affected areas. These visual explanations serve two crucial purposes: they provide transparency for the decision-making process and offer a layer of validation that the model is not relying on irrelevant features or background artifacts. For domain experts and farmers, this interpretability builds trust and supports adoption, as visual evidence aligns with agronomic knowledge of disease symptoms. By integrating Grad-CAM into our analysis, we not only demonstrate the robustness of our model but also make a significant step toward explainable and trustworthy AI in agricultural diagnostics.

## Discussion

Cotton crop classification is a crucial aspect of precision agriculture, enabling accurate yield estimation, disease detection, and optimal resource allocation. However, researchers faced several challenges that hindered the development of robust cotton crop classification systems. Firstly, data scarcity and imbalance pose a significant hurdle, with limited availability of high-quality data and disproportionate representation of certain crop classes. Secondly, extracting comprehensive features from cotton crop images is challenging due to their complex and variable nature. Hence, achieving accurate classification across seven cotton crop classes remains difficult due to the inherent similarity between some classes.

To further verify the generalization capability of the proposed model and address concerns related to potential overfitting due to synthetic data, we performed external validation using a publicly available cotton leaf disease dataset [76], which was completely unseen during training. Despite no prior exposure to this dataset, the proposed model achieved a high accuracy of 0.964, which is competitive with other recent methods that used this same dataset for both training and testing. This external validation reinforces the model's robustness and confirms its ability to generalize effectively to real-world data with different image resolutions, disease severities, and environmental backgrounds.

The critical examination of results reveals that the proposed method outperforms several notable methods, including ResNet152V2 [9], Modified Classifier Logic for Crop Monitoring [12], and SSADN-PLDDC technique with ELM [10], all of which were designed for datasets with fewer classes. The proposed method demonstrates superior performance compared to methods utilizing EfficientNet and MobileNet [16], Convolutional Neural Networks [2,33], Binary logistic regression [1], DNN inception-V3 pre-trained mode [35], K-mean clustering and SVM [36], KMSEG classifier [37], MLR, MLRb, SVM, RFT [38], Transfer learning [69] with GoogleNet CNN [39,40]. Feature Selection and classification using SVM [41]. Confirming that the best-performing classifier is SVM, with the MobileNetV2 feature set. This in-depth review highlights how well the suggested approach works and its ability to scale and adapt as it deals with an increasing variety of categories. The accurate results have been achieved due to an ensemble learning method that combines StyleGANs, VGG16, and MobileNetV2 within the StackNet framework. This approach takes advantage of automated feature extraction, concatenates the features, passes it to three different classifiers, and performs ensemble classification using StackNet.

Together with strong results on the PlantVillage [59] and Roboflow [60] datasets, the model's competitive performance on unseen data highlights its potential for deployment in real-time cotton disease classification applications.

### Key findings and potential significance

The proposed ensemble method, trained with categorical cross-entropy loss, RMSProp optimizer, and softmax activation, achieves an accuracy of 97%. We introduce a novel method for disease detection that helps in dealing with real-world challenges such as varying disease symptoms, occlusion from overlapping leaves, and diversity in leaves in terms of shape, size, and texture. A novel data augmentation technique has been developed to address the issue of data scarcity and class imbalance. Feature level fusion of multi-convolutional neural combined with ensemble classification techniques is a novel approach to improve the accuracy of disease detection with an increased number of disease classes. This method combines the strengths of convolutional neural networks for feature extraction and classification. As a result, it accurately identifies six specific diseases and a healthy class in cotton crops. Moreover, the robustness of the model was demonstrated through external testing on a completely unseen, publicly available dataset, where it achieved a high classification accuracy. This result highlights the model's strong generalization capability, suggesting its potential for effective deployment in real-world scenarios beyond controlled training environments.

### Limitations and future directions

The primary limitation of this study is the possibility of overfitting due to the controlled environment in which the data was collected and the models were trained. This controlled setting may not accurately reflect the variability and complexity of real-world field conditions, leading to lower performance when the models are applied in ream time scenarios. Moreover, a small dataset restricts the system's effectiveness and its ability to generalize well across diverse scenarios. To partially mitigate this limitation, we performed external validation using a completely unseen, real-world cotton leaf disease dataset. The model achieved a competitive accuracy of 0.964, despite not being exposed to this dataset during the training or validation phase. This indicates a strong generalization capability of the proposed model and supports its robustness against overfitting, even when applied to new field data from different domains.

While the external validation shows encouraging signs of real-world applicability, further improvements are necessary to ensure consistent performance in diverse and uncontrolled environments. Future research must focus on real-time data acquisition and pre-processing to increase the robustness of model in real environments [74]. Improving model interpretability and transparency will ensure AI decisions are understandable to farmers and agricultural professionals. Developing user-friendly interfaces and mobile applications will aid practical field use. Expanding the dataset with diverse images of diseased leaves will improve generalization and accuracy. Advanced algorithms for precise disease coverage localization will optimize pesticide usage, while refined segmentation techniques will better handle overlapping symptoms of multiple diseases on a single leaf.

## Conclusion

We have addressed three problems: data scarcity and imbalance, comprehensive feature set extraction, and accurate classification of seven classes of cotton crops. It has been observed that conventional data augmentation techniques are insufficient to address the problem of imbalance and scarcity. Therefore, we propose a customized StyleGANs network for data

augmentation and conventional methods. Similarly, a comprehensive feature set has been obtained by removing the fully connected VGG16 and MobileNet v2 layers and concatenating the features obtained from these two models. For accurate classification, we propose an ensemble classifier based on StackNet with SVM, LSTM, and Random Forest as base classifiers, and the the output is used in meta-learning. The proposed method has been applied to public datasets to validate the accuracy. It outperforms the existing methods of cotton crop disease detection in terms of both accuracy and number of classes. To further demonstrate the generalizability and robustness of the proposed model, we conducted external validation using a completely unseen publicly available dataset. Our methodology show results comparable to other state-of-the-art models, which affirms the ability of the model to handle diverse data in the real world and validates its potential for reliable use in practical agricultural applications.

## Author contributions

**Conceptualization:** Syed Muhammad Usman.

**Formal analysis:** Afira Aslam, Syed Muhammad Usman, Muhammad Zubair, Amanullah Yasin, Muhammad Owais.

**Funding acquisition:** Irfan Hussain.

**Investigation:** Muhammad Zubair, Amanullah Yasin.

**Methodology:** Afira Aslam, Syed Muhammad Usman, Amanullah Yasin, Muhammad Owais.

**Project administration:** Irfan Hussain.

**Resources:** Irfan Hussain.

**Supervision:** Muhammad Owais, Irfan Hussain.

**Validation:** Afira Aslam, Muhammad Zubair, Amanullah Yasin, Muhammad Owais.

**Visualization:** Afira Aslam, Muhammad Zubair.

**Writing – original draft:** Afira Aslam.

**Writing – review & editing:** Syed Muhammad Usman.

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
