## [Decision Letter · Decision Letter 0]

20 May 2024

PONE-D-24-09970Multi-Convolutional Neural Networks for Cotton Disease Detection Using Synergistic Deep Learning ParadigmPLOS ONE

Dear Dr. Owais,

Thank you for submitting your manuscript to PLOS ONE. After careful consideration, we feel that it has merit but does not fully meet PLOS ONE’s publication criteria as it currently stands. Therefore, we invite you to submit a revised version of the manuscript that addresses the points raised during the review process.

We look forward to receiving your revised manuscript.

Kind regards,

Jyotir Moy Chatterjee

Academic Editor

PLOS ONE

Journal Requirements:

"This work was supported by KUCARS, Department of Mechanical and Nuclear Engineering, Khalifa University under Award numbers FSU-2021-019 and RC1-2018-KUCARS."

"This work was supported by KUCARS, Department of Mechanical and Nuclear Engineering, Khalifa University under Award numbers FSU-2021-019 and RC1-2018-KUCARS."

"The author(s) received no specific funding for this work"

7. We note that your Data Availability Statement is currently as follows: All relevant data are within the manuscript and its Supporting Information files.

8. PLOS requires an ORCID iD for the corresponding author in Editorial Manager on papers submitted after December 6th, 2016. Please ensure that you have an ORCID iD and that it is validated in Editorial Manager. To do this, go to ‘Update my Information’ (in the upper left-hand corner of the main menu), and click on the Fetch/Validate link next to the ORCID field. This will take you to the ORCID site and allow you to create a new iD or authenticate a pre-existing iD in Editorial Manager. Please see the following video for instructions on linking an ORCID iD to your Editorial Manager account: https://www.youtube.com/watch?v=_xcclfuvtxQ

9. Please ensure that you refer to Figure 12 in your text as, if accepted, production will need this reference to link the reader to the figure.

10. We note you have included a table to which you do not refer in the text of your manuscript. Please ensure that you refer to Table 5 in your text; if accepted, production will need this reference to link the reader to the Table.

**Additional Editor Comments:**

Major revision

Reviewers' comments:

Reviewer's Responses to Questions

**Comments to the Author**

1. Is the manuscript technically sound, and do the data support the conclusions?

Reviewer #1: Yes

Reviewer #2: Yes

2. Has the statistical analysis been performed appropriately and rigorously? 

Reviewer #1: Yes

Reviewer #2: Yes

3. Have the authors made all data underlying the findings in their manuscript fully available?

Reviewer #1: No

Reviewer #2: Yes

4. Is the manuscript presented in an intelligible fashion and written in standard English?

Reviewer #1: Yes

Reviewer #2: Yes

5. Review Comments to the Author

Reviewer #1: Authors of this manuscript addressed the problem of disease detection in cotton crop and provide the solution to detect the disease by using deep learning method.

This is good piece of work but it require following corrections

1. The overall quality of the paper in general, and the implementation and presentation aspect need much improvement

2. The novelty of this research is not well explained

3. The statement “A robust system capable of addressing real-life challenges, such as diverse symptoms of diseases, occlusion, and variations in leaves” what is meaning by real life challenges is not clear.

4. Classification could be done by using SVM and random forest but LSTM is used for sequential data why all these are used collectively is not clearly explained.

5. How the stylGANs approach is addressing the problem of Data scarcity is not clear

6. Authors claimed that their method is outperforming than the existing methods but the comparison is not clear

Reviewer #2: 1. The author implement state of art method like UNet or UNet++ for segmentation.

2.Fig4 should be application specific relevant to the proposed work...it looks as general figure.

3. Table 7 is not clear, the values represent what?

4. Discuss the related work in such a manner that it should highlight the

method, results, advantages, and limitations. Add 8-10 more literature

and discuss it properly. The final review analysis should be added at the

end of this section. It will improve the readability of the paper. Add

papers from 2021-2023, also

Discussion section should contain the result impact, comparative

study analysis and the overall analysis of the result.

5. limitations and future scope need to be included

6. PLOS authors have the option to publish the peer review history of their article (what does this mean?). If published, this will include your full peer review and any attached files.

Reviewer #1: No

Reviewer #2: No

---

## [Author Response · Author response to Decision Letter 1]

28 Jun 2024

< Editor Comments >

Dear Editor,

We appreciate your review of our submitted manuscript titled “Multi-Convolutional Neural Networks for Cotton Disease Detection Using Ensemble Learning” in the PLOS One journal.

We have thoroughly revised our paper according to the “Journal Requirements” and the comments provided by the “Reviewers”. All the revisions are highlighted in red and blue text in the revised manuscript. We have also included a rebuttal to address the changes made in response to each reviewer's comments.

Best Regards,

Muhammad Owais (Ph.D.)

Corresponding author

muhammad.owais@ku.ac.ae

Contanct#: +971-58-135-8344

< Journal Requirements >

< Comment 1 >

< Response >

We confirm that our manuscript meets PLOS ONE’s style requirements, including those for file naming.

< Comment 2 >

We suggest you thoroughly copyedit your manuscript for language usage, spelling, and grammar. If you do not know anyone who can help you do this, you may wish to consider employing a professional scientific editing service.

< Response >

Prof. Dr. Shehzad Khalid proofread the manuscript. Both clean copy and track changes are being submitted.

< Comment 3 >

Please note that PLOS ONE has specific guidelines on code sharing for submissions in which author-generated code underpins the findings in the manuscript. In these cases, all author-generated code must be made available without restrictions upon publication of the work. Please review our guidelines at https://journals.plos.org/plosone/s/materials-and-software-sharing#loc-sharing-code and ensure that your code is shared in a way that follows best practice and facilitates reproducibility and reuse.

< Response >

The authors agree to share the code to reproduce the results of this research.

< Comment 4 >

We note that the grant information you provided in the ‘Funding Information’ and ‘Financial Disclosure’ sections do not match.

< Response >

Corrected: “This work was supported by KUCARS, Department of Mechanical and Nuclear Engineering, Khalifa University under Award numbers FSU-2021-019 and RC1-2018-KUCARS.”

< Comment 5 >

Thank you for stating the following financial disclosure:

"This work was supported by KUCARS, Department of Mechanical and Nuclear Engineering, Khalifa University under Award numbers FSU-2021-019 and RC1-2018-KUCARS."

< Response >

We amended the role of funder statement and also included in cover letter in yellow highlighted text. “Role of Funders: The funders oversaw this research and helped in preparing the manuscript.”

< Comment 6 >

Thank you for stating the following in the Acknowledgments Section of your manuscript:

"This work was supported by KUCARS, Department of Mechanical and Nuclear Engineering, Khalifa University under Award numbers FSU-2021-019 and RC1-2018-KUCARS."

"The author(s) received no specific funding for this work"

< Response >

We have renamed the “Funding and Acknowledgement” section to “Funding Statement” in the revised manuscript and updated the funding statement as follows:

Previous statement:

Revised statement:

This work was supported by KUCARS, Department of Mechanical and Nuclear Engineering, Khalifa University under Award numbers FSU-2021-019 and RC1-2018-KUCARS.

< Comment 7 >

We note that your Data Availability Statement is currently as follows: All relevant data are within the manuscript and its Supporting Information files.

< Response >

In this study, we used publicly available open datasets. These publicly available datasets links are provided below:

Data source links:

https://www.kaggle.com/datasets/seroshkarim/cotton-leaf-disease-dataset

https://data.mendeley.com/datasets/6hm6pc3y43/2

https://universe.roboflow.com/national-college-of-ireland/cotton-plant-disease-prediction-igthk/dataset/3

https://universe.roboflow.com/roboflow-100/cotton-plant-disease/dataset/2

There are no such graphs that used mean, standard deviation, etc., values to plot. However, to reproduce the research work authors have included the details on the data augmentation (Table 3), model details (Table 4, 5, and 8).

< Comment 8 >

PLOS requires an ORCID iD for the corresponding author in Editorial Manager on papers submitted after December 6th, 2016. Please ensure that you have an ORCID iD and that it is validated in Editorial Manager. To do this, go to ‘Update my Information’ (in the upper left-hand corner of the main menu), and click on the Fetch/Validate link next to the ORCID field. This will take you to the ORCID site and allow you to create a new iD or authenticate a pre-existing iD in Editorial Manager. Please see the following video for instructions on linking an ORCID iD to your Editorial Manager account: https://www.youtube.com/watch?v=_xcclfuvtxQ

< Response >

Thanks for your update. The corresponding author has linked his ORCID with the editorial manager.

https://orcid.org/0000-0001-7679-081X

< Comment 9 >

Please ensure that you refer to Figure 12 in your text as, if accepted, production will need this reference to link the reader to the figure.

< Response >

Figure 12 has been referenced in the text and updated in the revised manuscript. Please see 2nd paragraph of “Results” section.

< Comment 10 >

We note you have included a table to which you do not refer in the text of your manuscript. Please ensure that you refer to Table 5 in your text; if accepted, production will need this reference to link the reader to the Table.

< Response >

Authors would like to thank the editor and have referenced the Table 5 in the text.

Please see the “Proposed Methodology” section, the paragraph starting with the following statement:

Table 5 provides detail of parameters of MobileNet V2. 

< Reviewer 1 >

< General Comment >

Authors of this manuscript addressed the problem of disease detection in cotton crop and provide the solution to detect the disease by using deep learning method. This is good piece of work but it require following corrections.

< Response >

Thanks for your positive response and valuable comments. We have thoroughly revised our paper according to the comments. All the revisions are highlighted in red and blue text in the revised manuscript.

< Comment 1 >

The overall quality of the paper in general, and the implementation and presentation aspect need much improvement.

< Response >

We have thoroughly revised our paper by addressing the implementation and result presentation to improve the overall quality of the paper.

< Comment 2 >

The novelty of this research is not well explained

< Response >

We apologize for the confusion. We have thoroughly revised the contribution and discussion sections of our paper to highlight the novelty of this research as follows:

Novelty of the study:

We proposed a comprehensive system designed to effectively manage and overcome these challenges, ensuring reliable and accurate performance in diverse agricultural environments. Contributions of this research to the body of knowledge include:

• Developed a deep ensemble method to accurately classify various cotton diseases, addressing challenges of class imbalance and symptom variation through a new data augmentation scheme and feature fusion approach.

• Proposed a synthetic data augmentation scheme using a customized StyleGAN along with conventional methods (scaling, rotating, transforming, shearing, zooming) to tackle class imbalance.

• Introduced a dual-stage fusion scheme by combining features extracted from MobileNet and VGG16 to create a comprehensive feature vector, followed by a StackNet-based ensemble classifier integrating Long and Short Term Memory (LSTM), Support Vector Machine (SVM), and Random Forest for enhanced prediction accuracy.

• Achieved an average accuracy of 97% on a publicly available dataset, surpassing state-of-the-art techniques in identifying six diseases and a healthy class.

These modifications are included at the end of the “Introduction” section.

Key Findings and Potential Significance:

The proposed ensemble method, trained with categorical cross-entropy loss, RMSProp optimizer, and softmax activation, achieves a 97\% accuracy. This research introduces several novel aspects of precision agriculture, particularly in detecting and managing crop diseases and developing a robust method that deals with real-world challenges such as varying disease symptoms, occlusion from overlapping leaves, and diverse leaf characteristics in shape, size, and texture. A novel data augmentation technique has been developed to address the issue of data scarcity and class imbalance in agricultural datasets. Incorporating multi-convolutional neural networks combined with ensemble classification techniques is a novel approach to improve the accuracy of disease detection. This method combines the strengths of convolutional neural networks for feature extraction and classification. As a result, it accurately identifies six specific diseases and a healthy class in cotton crops.

The above paragraph highlighting the novelty of this study is also added as the last paragraph in the “Discussion” section.

< Comment 3 >

“A robust system capable of addressing real-life challenges, such as diverse symptoms of diseases, occlusion, and variations in leaves” what is meaning by real life challenges is not clear

< Response >

We apologize for the vague explanation. We revised these lines as follows:

A robust deep learning system is essential for addressing the practical challenges encountered in real-world agricultural settings. These challenges include diverse symptoms of diseases, such as varying sizes and colors of lesions, which can complicate accurate diagnosis. Additionally, occlusion, where leaves are partially covered by other leaves or objects, poses a significant obstacle to clear image capture and analysis. Variations in leaf appearance, including differences in shapes, sizes, and textures, further add to the complexity. To tackle these issues, we proposed a comprehensive system designed to effectively manage and overcome these challenges, ensuring reliable and accurate performance in diverse agricultural environments.

These lines are updated at the end of the “Introduction” section (before the 1st bullet point of the contribution part).

< Comment 4 >

Classification could be done by using SVM and random forest but LSTM is used for sequential data why all these are used collectively is not clearly explained.

< Response >

Thanks for your valuable comment. We added further details to the proposed methodology to explain the significance of LSTM, which is used along with SVM and RF as followas:

The use of LSTM, despite its typical application for sequential data, is beneficial as it captures temporal dependencies or patterns within the feature vectors that traditional classifiers like SVM and Random Forest might overlook. By integrating these diverse methods, the strength of LSTM in handling sequential patterns is leveraged alongside the robustness of Random Forest in managing non-linear relationships and the effectiveness of SVM in high-dimensional spaces, thereby enhancing overall model performance.

Please see the 2nd paragraph of the section “Proposed Methodology”.

< Comment 5 >

How the stylGANs approach is addressing the problem of Data scarcity is not clear.

< Response >

Explanation is added to clarify how styleGANs help with data scarcity as follows:

StyleGANs address the problem of data scarcity by generating high-quality synthetic images that are indistinguishable from real images. This increases the dataset size and ensures diversity and variety in the augmented data, thus enhancing the model's ability to generalize better.

Please see the 3rd paragraph of the section “Proposed Methodology”.

< Comment 6 >

Authors claimed that their method is outperforming than the existing methods but the comparison is not clear

< Response >

We apologize for the vague explanation. We have thoroughly revised the comparison table and highlighted the performance gains of the proposed method compared to existing methods as follows:

Comparison

Table 8 presents a comparative a

---

## [Decision Letter · Decision Letter 1]

8 Aug 2024

PONE-D-24-09970R1Multi-Convolutional Neural Networks for Cotton Disease Detection Using Synergistic Deep Learning ParadigmPLOS ONE

Dear Dr. Owais,

Thank you for submitting your manuscript to PLOS ONE. After careful consideration, we feel that it has merit but does not fully meet PLOS ONE’s publication criteria as it currently stands. Therefore, we invite you to submit a revised version of the manuscript that addresses the points raised during the review process.

We look forward to receiving your revised manuscript.

Kind regards,

Jyotir Moy Chatterjee

Academic Editor

PLOS ONE

Additional Editor Comments:

**Major Revision**

Reviewers' comments:

Reviewer's Responses to Questions

**Comments to the Author**

1. If the authors have adequately addressed your comments raised in a previous round of review and you feel that this manuscript is now acceptable for publication, you may indicate that here to bypass the “Comments to the Author” section, enter your conflict of interest statement in the “Confidential to Editor” section, and submit your "Accept" recommendation.

Reviewer #1: All comments have been addressed

Reviewer #3: All comments have been addressed

2. Is the manuscript technically sound, and do the data support the conclusions?

Reviewer #1: Yes

Reviewer #3: Partly

3. Has the statistical analysis been performed appropriately and rigorously? 

Reviewer #1: Yes

Reviewer #3: Yes

4. Have the authors made all data underlying the findings in their manuscript fully available?

Reviewer #1: Yes

Reviewer #3: No

5. Is the manuscript presented in an intelligible fashion and written in standard English?

Reviewer #1: Yes

Reviewer #3: Yes

6. Review Comments to the Author

Reviewer #1: The authors edited the manuscript according to the comments. Therefore accepted for the publication.

Reviewer #3: The authors have not been able to show the new contribution. This paper does not add anything new to knowledge as the authors dont state the major contributions of the work. There is no comparative analysis with the state-of-the-art methods.

7. PLOS authors have the option to publish the peer review history of their article (what does this mean?). If published, this will include your full peer review and any attached files.

Reviewer #1: No

Reviewer #3: No

---

## [Author Response · Author response to Decision Letter 2]

17 Aug 2024

< Reviewer 1 >

< Comment >

The authors edited the manuscript according to the comments. Therefore, accepted for the publication.

< Response >

We would like to thank the reviewer and appreciate your reviewing process of our manuscript.

< Reviewer 3 >

< Comment 1 >

The authors have not been able to show the new contribution. This paper does not add anything new to knowledge as the authors dont state the major contributions of the work.

< Response >

Authors would like to thank the reviewer for the valuable comment. We have highlighted the contributions and incorporated them in the Introduction section, lines 48-57 on page no. 3.

Action taken in the revised manuscript

• Developed a novel method for data augmentation by combining conventional methods and custom Style GANs to reduce class imbalance problems.

• Introduced feature-level fusion scheme by combining features extracted from MobileNet and VGG16 to create a comprehensive feature vector

• Proposed an ensemble classifier using StackNet by integrating Long Short Term Memory Units (LSTM), Support Vector Machine (SVM), and Random Forest to classify cotton crop diseases accurately.

• Achieved an average accuracy of 97\% on publicly available datasets, surpassing state-of-the-art techniques in classifying seven classes, including six diseases and a healthy class.

< Comment 2 >

There is no comparative analysis with the state-of-the-art methods.

< Response >

The authors would like to thank the reviewer for the valuable suggestion. We have added a comparison of the proposed method with existing state-of-the-art methods. Table 8 has been updated in the results section on page no. 17 and described in the text.

Action taken in the revised manuscript

Table 8 presents a comparative analysis of the proposed model versus existing methods for cotton disease detection, focusing on their parameters (#Par in millions) and accuracy. The proposed ensemble model achieves the highest accuracy of 0.97, indicating a 6% performance gain over the next best models. The second-best models [50, 54], with accuracies of 0.91 and 0.90, include MobileNetV2 with a learning scheduler (0.91, 3.5M), MobileNetV2 with more dense layers (0.90, 3.5M), and EfficientNet (0.90, 5.3M). The third-best models [50, 51], with an accuracy of 0.88, include MobileNetV2 (3.5M), MobileNetV2 unfreezing initial layers (3.5M) and Xception (22.9M). Despite the proposed method’s high parameter count (143.4M), its 6% performance gain justifies the complexity, significantly outperforming models like MobileNetV2 (3.5M) and DenseNet121 (8.0M), which achieve relatively high accuracy. This highlights the effectiveness of our ensemble model in achieving superior accuracy, balancing parameter size and performance gain, and demonstrating the model’s efficacy in cotton disease detection applications.

---

## [Decision Letter · Decision Letter 2]

27 Dec 2024

PONE-D-24-09970R2Multi-Convolutional Neural Networks for Cotton Disease Detection Using Synergistic Deep Learning ParadigmPLOS ONE

Dear Dr. Owais,

Thank you for submitting your manuscript to PLOS ONE. After careful consideration, we feel that it has merit but does not fully meet PLOS ONE’s publication criteria as it currently stands. Therefore, we invite you to submit a revised version of the manuscript that addresses the points raised during the review process.

**ACADEMIC EDITOR: Minor Revision**==============================

We look forward to receiving your revised manuscript.

Kind regards,

Jyotir Moy Chatterjee

Academic Editor

PLOS ONE

Journal Requirements:

Reviewers' comments:

Reviewer's Responses to Questions

**Comments to the Author**

1. If the authors have adequately addressed your comments raised in a previous round of review and you feel that this manuscript is now acceptable for publication, you may indicate that here to bypass the “Comments to the Author” section, enter your conflict of interest statement in the “Confidential to Editor” section, and submit your "Accept" recommendation.

Reviewer #4: All comments have been addressed

Reviewer #5: All comments have been addressed

2. Is the manuscript technically sound, and do the data support the conclusions?

Reviewer #4: Yes

Reviewer #5: Partly

3. Has the statistical analysis been performed appropriately and rigorously? 

Reviewer #4: Yes

Reviewer #5: Yes

4. Have the authors made all data underlying the findings in their manuscript fully available?

Reviewer #4: Yes

Reviewer #5: Yes

5. Is the manuscript presented in an intelligible fashion and written in standard English?

Reviewer #4: Yes

Reviewer #5: Yes

6. Review Comments to the Author

Reviewer #4: (No Response)

Reviewer #5: Comments:

Table 8 - Major Revision Needed: Table 8 presents important findings that compare various methods and their associated accuracies. However, there is significant ambiguity in the dataset used and some discrepancies with the reported accuracies.

Dataset Consistency: It is crucial to clarify whether all the studies referenced in Table 8 are based on the same dataset. The manuscript does not specify whether these methods are evaluated using identical datasets or if different datasets were used across the studies. This is essential for interpreting the accuracy comparisons accurately. I recommend adding a note to Table 8 or in the figure legend specifying whether the methods are evaluated on the same dataset, and if not, providing details about the datasets used for each study.

Inconsistencies in Reported Accuracy:

The manuscript mentions that the accuracy of the method from Reference 53 is 86%, but the published version of Reference 53 states that the accuracy of this method is 96.46%. This discrepancy needs to be addressed. Please confirm the accuracy values, and if there was an error in reporting, correct it in both the manuscript and Table 8.

Similarly, the manuscript states that the accuracy for the method from Reference 50 is 0.88, but the published copy of Reference 50 does not report this value as accuracy. Instead, 0.88 refers to the Area Overlap Measure (AOM), which is a different metric. The manuscript should correct this mistake and clarify that 0.88 corresponds to the AOM, not accuracy.

7. PLOS authors have the option to publish the peer review history of their article (what does this mean?). If published, this will include your full peer review and any attached files.

Reviewer #4: No

Reviewer #5: No

---

## [Author Response · Author response to Decision Letter 3]

5 Feb 2025

REVIEWER – 5

Comments: Table 8 - Major Revision Needed:

Table 8 presents important findings that compare various methods and their associated accuracies. However, there is significant ambiguity in the dataset used and some discrepancies with the reported accuracies.

Dataset Consistency: It is crucial to clarify whether all the studies referenced in Table 8 are based on the same dataset. The manuscript does not specify whether these methods are evaluated using identical datasets or if different datasets were used across the studies. This is essential for interpreting the accuracy comparisons accurately. I recommend adding a note to Table 8 or in the figure legend specifying whether the methods are evaluated on the same dataset, and if not, providing details about the datasets used for each study.

Inconsistencies in Reported Accuracy: The manuscript mentions that the accuracy of the method from Reference 53 is 86%, but the published version of Reference 53 states that the accuracy of this method is 96.46%. This discrepancy needs to be addressed. Please confirm the accuracy values, and if there was an error in reporting, correct it in both the manuscript and Table 8. Similarly, the manuscript states that the accuracy for the method from Reference 50 is 0.88, but the published copy of Reference 50 does not report this value as accuracy. Instead, 0.88 refers to the Area Overlap Measure (AOM), which is a different metric. The manuscript should correct this mistake and clarify that 0.88 corresponds to the AOM, not accuracy.

Response:

The authors would like to thank the reviewer for the valuable comment.

We appreciate the reviewer's valuable feedback regarding the dataset consistency and reported accuracy values in Table 8. Based on the suggestions provided, we have made the necessary clarifications and corrections to enhance the accuracy and transparency of the reported findings.

Dataset Consistency Clarification:

We acknowledge the reviewer's concern regarding dataset consistency across the referenced studies. As suggested, we have explicitly added a new column to Table 8, specifying the datasets used in each study along with their respective references. This addition ensures clarity regarding the sources of data and allows for a more accurate interpretation of the accuracy comparisons. Furthermore, we have updated the table caption to explicitly state that the studies utilize different datasets for either binary or multiclass classification. These modifications directly address the reviewer's request for transparency in dataset usage.

Corrections to Reported Accuracy Values:

We have carefully reviewed and rectified any discrepancies in the reported accuracy values. Specifically:

• The accuracy value for Reference 53 was previously incorrect and has now been corrected to match the published value (0.96).

• The value for Reference 50 is corrected with (0.91) for MobileNet V2 and (0.98) with added dense layers.

• In addition to these corrections, we have thoroughly counterchecked the accuracy and parameter values for all other studies in the table to ensure accuracy and consistency. Any discrepancies identified during this process have been corrected accordingly.

Action: Please find in the attached response file

---

## [Decision Letter · Decision Letter 3]

10 Mar 2025

PONE-D-24-09970R3Multi-Convolutional Neural Networks for Cotton Disease Detection Using Synergistic Deep Learning ParadigmPLOS ONE

Dear Dr. Owais,

Thank you for submitting your manuscript to PLOS ONE. After careful consideration, we feel that it has merit but does not fully meet PLOS ONE’s publication criteria as it currently stands. Therefore, we invite you to submit a revised version of the manuscript that addresses the points raised during the review process.

**ACADEMIC EDITOR: **

We look forward to receiving your revised manuscript.

Kind regards,

Jyotir Moy Chatterjee

Academic Editor

PLOS ONE

Additional Editor Comments:

1. The study lacks clarity on whether all referenced methods in Table 8 are evaluated using the same dataset. This makes it difficult to accurately compare accuracy metrics across different approaches.

2. Discrepancies exist in reported accuracy values for references. For example, the manuscript previously stated that Reference 53 had an accuracy of 86%, but the published value was 96.46%. Similarly, Reference 50's accuracy was misrepresented.

3. The model is trained in a controlled setting, which may not accurately reflect real-world variability, including lighting conditions, environmental noise, and different imaging devices. This could lead to reduced generalizability when deployed in practical scenarios.

4. Although the study attempts to address class imbalance through synthetic data generation (StyleGAN and augmentation), reliance on synthetic data might introduce biases that do not accurately represent real-world variations in disease symptoms.

5. The reported 97% accuracy suggests a potential overfitting issue, especially given that the dataset is relatively small and augmented artificially. More robust validation techniques, such as external dataset testing, would strengthen the claims.

6. The study does not provide sufficient interpretability techniques (e.g., Grad-CAM, SHAP) to understand why the model makes certain predictions. Explainable AI methods could enhance trust in the system for agricultural experts and farmers.

Reviewers' comments:

Reviewer's Responses to Questions

**Comments to the Author**

1. If the authors have adequately addressed your comments raised in a previous round of review and you feel that this manuscript is now acceptable for publication, you may indicate that here to bypass the “Comments to the Author” section, enter your conflict of interest statement in the “Confidential to Editor” section, and submit your "Accept" recommendation.

Reviewer #6: (No Response)

2. Is the manuscript technically sound, and do the data support the conclusions?

Reviewer #6: Yes

3. Has the statistical analysis been performed appropriately and rigorously? 

Reviewer #6: Yes

4. Have the authors made all data underlying the findings in their manuscript fully available?

Reviewer #6: Yes

5. Is the manuscript presented in an intelligible fashion and written in standard English?

Reviewer #6: Yes

6. Review Comments to the Author

Reviewer #6: I found this to be a solid paper on cotton disease detection. The authors' approach of combining features from MobileNet and VGG16, then using an ensemble classifier (StackNet with LSTM, SVM, and Random Forest) is creative and effective.

What works well:

Their method addresses real challenges in field conditions, particularly variation in disease symptoms

The StyleGAN for synthetic data generation is a clever solution to class imbalance

Using the Segment Anything Model for disease area quantification adds practical value

A few suggestions that might strengthen the paper:

I'd like to see more about inference time and computational requirements, since that matters for field deployment

While they do compare parameter counts (143.4M is quite large), some discussion about deployment on resource-limited devices would be helpful

More testing on challenging field conditions (varied lighting, partial leaf occlusion) would better demonstrate robustness

The corrections they made to Table 8 in response to Reviewer 5 have addressed those concerns appropriately.

Overall, this represents a meaningful contribution that advances cotton disease detection methods.

7. PLOS authors have the option to publish the peer review history of their article (what does this mean?). If published, this will include your full peer review and any attached files.

Reviewer #6: No

---

## [Author Response · Author response to Decision Letter 4]

20 Apr 2025

Dear Editor,

We appreciate your reviewing process of our submitted manuscript titled “Multi-Convolutional Neural Networks for Cotton Disease Detection Using Ensemble Learning” in the PLOS One journal.

We have revised our paper according to the comments provided by the editor and reviewers. All added text is in Red, with removed text as (strike through) in the “Revised Manuscript with Track Changes”. Moreover, a “Response to Reviewer(s)” file is also provided to address the changes made in the revised paper against individual comments by the reviewers. In addition, a clean file “Manuscript” is also uploaded in separate.

Best regards

Editor Comments

Comment 1: The study lacks clarity on whether all referenced methods in Table 8 are evaluated using the same dataset. This makes it difficult to accurately compare accuracy metrics across different approaches.

Response: We appreciate the reviewer’s observation. In Table 8 of the original manuscript, we presented the results of prior studies primarily based on the "PlantVillage" dataset. However, our proposed model was evaluated on a combined dataset comprising both the "PlantVillage" and "Roboflow" datasets. We acknowledge that this discrepancy may have made cross-study comparisons unclear.

To address this, we have revised Table 8 to include recent studies that report performance on the "Roboflow" dataset as well. This ensures a more accurate and meaningful comparison. While two existing studies report slightly higher accuracy on the "PlantVillage" dataset alone, it is important to highlight that our model demonstrates robust performance across both datasets when combined, suggesting better generalization capability. Moreover, we have removed the result of one study which is shows results on a completely different dataset.

Our model’s strong performance on the combined datasets “PlantVillage” and “Roboflow” datasets is considered more challenging, further emphasizes its effectiveness and practical applicability.

Action: Please see the section “Comparison with State-of-the-Art methods” and Table 8 on page numbers 16-17 in the revised track changes manuscript.

Comment 2: Discrepancies exist in reported accuracy values for references. For example, the manuscript previously stated that Reference 53 had an accuracy of 86%, but the published value was 96.46%. Similarly, Reference 50's accuracy was misrepresented.

Response: We appreciate the reviewer for pointing out these discrepancies. Upon review, we found that Reference 53 was based on a different dataset that was not used in our model evaluation. To maintain a fair and consistent comparison, we have removed this reference from Table 8, as now reflected in the revised manuscript. Additionally, the accuracy value for Reference 50 has been corrected accordingly. We have also thoroughly verified all other referenced accuracy values to ensure consistency and accuracy across the manuscript.

Action: Please see Table 8 on page numbers 17 in the revised track changes manuscript.

Comment 3: The model is trained in a controlled setting, which may not accurately reflect real-world variability, including lighting conditions, environmental noise, and different imaging devices. This could lead to reduced generalizability when deployed in practical scenarios.

Response: We acknowledge the reviewer’s concern and have addressed this by evaluating our model on a combined dataset comprising PlantVillage and Roboflow, which differ significantly in terms of image quality, lighting conditions, background complexity, and noise. This combination introduces higher data variability and better reflects real-world scenarios. Unlike single-dataset studies, our approach provides a more rigorous test of generalization. Notably, our model performed strongly on the more challenging Roboflow dataset, demonstrating its robustness and applicability in diverse practical settings.

Action: Please see the added text on page number 17 in the revised track changes manuscript.

Comment 4: Although the study attempts to address class imbalance through synthetic data generation (StyleGAN and augmentation), reliance on synthetic data might introduce biases that do not accurately represent real-world variations in disease symptoms.

Response: We appreciate the reviewer’s valuable observation. While synthetic data generation using StyleGAN and augmentation was used to address class imbalance, we recognize the potential risk of introducing biases that may not fully capture the complexity of real-world disease manifestations. To mitigate this, we ensured that the synthetic samples were only used during the training phase and not in validation or testing, which were conducted exclusively on real images from the combined PlantVillage and Roboflow datasets.

Action: See 2nd paragraph of the “Proposed Methodology” on page number 8 in the revised track changes manuscript.

Comment 5: The reported 97% accuracy suggests a potential overfitting issue, especially given that the dataset is relatively small and augmented artificially. More robust validation techniques, such as external dataset testing, would strengthen the claims.

Response: We thank the reviewer for raising this important concern. To address the potential overfitting issue, we conducted an external validation using a completely unseen, publicly available cotton leaf disease dataset [74]. Unlike prior studies that used parts of this dataset for both training and testing, our model was evaluated solely on the test set without any prior exposure. Despite this, our model achieved a competitive accuracy of 96.4%, confirming its generalization ability. These results, combined with strong performance on diverse datasets like PlantVillage and Roboflow, demonstrate the robustness and real-world applicability of our model. We have included this external validation and discussion in the revised manuscript (see Section "External Validation Testing" and Table 9). Moreover, the sections “Discussion”, “Key Findings and Potential Significance”, “Limitations and Future Directions”, and “Conclusion” are also updated accordingly.

Action: See section “External Dataset Testing” on page number 18, Table 9 on page 19, and “Discussion” section on page 19 and 21, “Key Findings and Potential Significance” section on page 21, “Limitations and Future Directions” on page 21-22, and “Conclusion” section on page 22 in the revised track changes manuscript.

Discussion

Key Findings and Potential Significance

Limitations and Future Directions

Conclusion

Comment 6: The study does not provide sufficient interpretability techniques (e.g., Grad-CAM, SHAP) to understand why the model makes certain predictions. Explainable AI methods could enhance trust in the system for agricultural experts and farmers.

Response: We appreciate the reviewer’s valuable suggestion. In the revised manuscript, we have incorporated the Explainable AI technique Grad-CAM to provide visual explanations for the model's predictions. A new section titled “Visual Explanation of Model Decisions” has been added, along with Figure 9, which presents Grad-CAM visualizations for VGG-16, MobileNet V2, StackNet, and our proposed ensemble model. These visualizations highlight the discriminative regions used by each model, thereby improving transparency and helping end-users understand and trust the model’s decision-making process.

Action: See section “Visual Explanation of Model Decisions” on page 19 and Figure 9 on page 20 in the revised track changes manuscript.

---

## [Editor Report · Decision Letter 4]

23 Apr 2025

Multi-Convolutional Neural Networks for Cotton Disease Detection Using Synergistic Deep Learning Paradigm

PONE-D-24-09970R4

Dear Dr. Owais,

We’re pleased to inform you that your manuscript has been judged scientifically suitable for publication and will be formally accepted for publication once it meets all outstanding technical requirements.

Kind regards,

Jyotir Moy Chatterjee

Academic Editor

PLOS ONE
---

## [Editor Report · Acceptance letter]

PONE-D-24-09970R4

PLOS ONE

Dear Dr. Owais,

I'm pleased to inform you that your manuscript has been deemed suitable for publication in PLOS ONE. Congratulations! Your manuscript is now being handed over to our production team.

Kind regards,

on behalf of

Mr. Jyotir Moy Chatterjee

Academic Editor

PLOS ONE